



# Estimating CH₄, CO₂, and CO emissions from coal mining and industrial activities in the Upper Silesian Coal Basin using an aircraft-based mass balance approach

Alina Fiehn[1], Julian Kostinek[1], Maximilian Eckl[1], Theresa Klausner[1], Michał Gałkowski[2,3], Jinxuan Chen[2], Christoph Gerbig[2], Thomas Röckmann[4], Hossein Maazallahi[4], Martina Schmidt[5], Piotr Korbeń[5], Jarosław Nęcki[3], Pawel Jagoda[3], Norman Wildmann[1], Christian Mallaun[6], Rostyslav Bun[7,8], Anna-Leah Nickl[1], Patrick Jöckel[1], Andreas Fix[1], Anke Roiger[1]

[1] Deutsches Zentrum für Luft- und Raumfahrt (DLR), Institut für Physik der Atmosphäre, Oberpfaffenhofen, Germany
[2] Max-Planck-Institut für Biogeochemie (MPI-BGC), Jena, Germany
[3] Faculty of Physics and Applied Computer Science, AGH University of Science and Technology, Kraków, Poland
[4] Institute for Marine and Atmospheric research Utrecht, Utrecht University, Utrecht, The Netherlands
[5] Institute of Environmental Physics, University of Heidelberg, Heidelberg, Germany
[6] Deutsches Zentrum für Luft- und Raumfahrt (DLR), Flugexperimente, Oberpfaffenhofen, Germany
[7] Department of Applied Mathematics, Lviv Polytechnic National University, Ukraine
[8] Faculty of Applied Sciences, WSB University, Dąbrowa Górnicza, Poland

*Correspondence to*: Alina Fiehn (alina.fiehn@dlr.de)

**Abstract.** A severe reduction of greenhouse gas emissions is necessary to reach the objectives of the Paris Agreement. The implementation and continuous evaluation of mitigation measures requires regular independent information on emissions of the two main anthropogenic greenhouse gases, carbon dioxide ($CO_2$) and methane ($CH_4$). Our aim is to employ an observation-based method to determine regional-scale greenhouse gas emission estimates with high accuracy. We use aircraft- and ground-based in situ observations of $CH_4$, $CO_2$, carbon monoxide (CO), and wind speed from two research flights over the Upper Silesian Coal Basin (USCB), Poland, in summer 2018. The flights were performed as a part of the Carbon Dioxide and Methane (CoMet) mission above this European $CH_4$ emission hot spot region. A kriging algorithm interpolates the observed concentrations between the downwind transects of the trace gas plume and then the mass flux through this plane is calculated. Finally, statistic and systematic uncertainties are calculated from measurement uncertainties and through several sensitivity tests, respectively.

For the two selected flights, the in situ derived annual $CH_4$ emission estimates are $13.8 \pm 3.6$ kg/s and $15.1 \pm 3.0$ kg/s, which is well within the range of emission inventories. The regional emission estimates of $CO_2$, which were determined to be $1.21 \pm 0.72$ t/s and $1.12 \pm 0.37$ t/s, are in the lower range of emission inventories. CO mass balance emissions of $10.1 \pm 3.2$ kg/s and $10.7 \pm 2.9$ kg/s for the USCB are slightly higher than the emission inventory values. The $CH_4$ emission estimate has a relative error of 21-26%, the $CO_2$ estimate of 33-60%, and the CO estimate of 27-32%. These errors mainly result from the uncertainty of atmospheric background mole fractions and the changing planetary boundary layer height during the morning flight. In the case of $CO_2$, biospheric fluxes also add to the uncertainty and hamper the assessment of emission inventories.



These emission estimates characterize the USCB and help to verify emission inventories and develop climate mitigation

strategies.

## 1 Introduction

One of the main objectives of the Paris Agreement is to keep the global temperature rise well below 2°C compared to pre-industrial levels (UNFCCC, 2015). This ambitious goal can only be reached by a severe reduction of greenhouse gas emissions. The development of efficient mitigation strategies and the implementation and management of long-term policies

requires consistent, reliable, and timely information on emissions of the two main anthropogenic greenhouse gases, carbon dioxide ($CO_2$) and methane ($CH_4$). Carbon monoxide (CO) can be used as an additional tracer for comparison with emission inventories and as proxy for $CO_2$ from fossil fuel combustion. It is produced from the incomplete combustion of fossil fuels and biomass and reacts with the hydroxyl radical (OH), thus affects the main sink of $CH_4$.

The globally averaged atmospheric abundances of $CO_2$ and $CH_4$ have increased by 47% to $407.8 \pm 0.1$ ppm and by 159% to

$1869 \pm 2$ ppb, respectively, in the period 1750 to 2018 (WMO, 2019). The relative contribution of individual sources and sinks to atmospheric $CH_4$ is still highly uncertain and the factors that affect these sources and sinks are not fully understood (Saunois et al., 2019). After a period of stable mole fractions since 2000, the atmospheric abundance of $CH_4$ has started to increase again in 2007, and after 2014 the increase intensified yet again (Nisbet et al., 2014; Nisbet et al., 2016). The reason for this increased growth is currently investigated in several studies, which partly contradict each other by discussing

biogenic sources, fossil fuel emissions and/or a decrease in the OH sink (Hausmann et al., 2016; Schaefer et al., 2016; Saunois et al., 2017; Turner et al., 2017; Worden et al., 2017; Nisbet et al., 2019).

Atmospheric emission inventories for trace species are usually based on *bottom-up* data-based approaches. Here, emissions for individual facilities, sectors, or sources are compiled into a comprehensive database. If direct emission data is not available, they are often calculated using activity data, like the mass of coal extracted, together with emission factors. For

Annex I countries, sector specific emissions of greenhouse gases have to be reported annually under the United Nations Framework Convention on Climate Change (UNFCC). Other countries are encouraged to report national totals of emissions. Bottom-up inventories can thus include single-source emissions, national totals, or can be disaggregated on different spatial scales. These gridded emission inventories commonly use national emission totals and distribute them across each country using proxy data like population density or single facility locations. This method is used to compile emission inventories,

which are used e.g. in climate projections. The neglect of regional differences and the uncertainties in the proxy data and emission factors introduce high uncertainties into the emission inventories at grid cell level (Janssens-Maenhout et al., 2019). Without accurate emission estimates it is challenging to create reliable future climate projections and develop efficient mitigation strategies.

Therefore, there is a strong need for an independent and objective verification of emissions from individual sources or source

regions based on atmospheric observations, usually referred to as *top-down* approaches. Top-down studies based on satellite data provide information on global and regional scales. For methane, emission quantification of individual sources has





recently been demonstrated on very large point sources (Pandey et al., 2019; Varon et al., 2019), but quantification of smaller sources is still difficult. Here, airborne measurements reveal more detailed insights on smaller scales, because in situ measurements allow the study of emission sources with high spatial resolution and accuracy. High precision measurements

of atmospheric concentration can be used for the top-down estimation of emissions from specific regions or sectors using atmospheric inversion models (Gurney et al., 2002; Thompson et al., 2014; Bergamaschi et al., 2018), and for the validation of numerical models used to calculate atmospheric abundances based on bottom-up emission inventories (Krinner et al., 2005; O'Shea et al., 2014). Airborne measurements provide highly valuable data for an independent assessment of anthropogenic $CH_4$, $CO_2$ and CO emissions, because the majority of these emissions originate from a small fraction of the

globe, namely fossil fuel exploitation facilities, cities, and power plants. Airborne measurements have shown to be useful in emission assessment of anthropogenic emissions from several sectors, including landfills (Cambaliza, 2015; Krautwurst et al., 2017) and oil and gas production regions (Karion et al., 2015; Yuan et al., 2015; Alvarez et al., 2018; Barkley et al., 2019). Plant et al. (2019) and Ren et al. (2018) showed that North American cities emit more $CH_4$ than suspected, because of underestimation of natural gas leakage or lack of inclusion of end use emissions.

Aircraft top-down approaches can be used in several ways to obtain greenhouse gas flux estimates. One way is the mass balance approach, where the emissions are estimated from observed in situ mole fractions and wind speeds in the target region. Different flight patterns are used for mass balance studies: A single downwind flight transect in the approximate vertical center of the boundary layer (Karion et al., 2013) or several transects of the plume at the same height but different distances from the source (Turnbull et al., 2011) are sufficient in case of a well-mixed planetary boundary layer (PBL). A

better understanding of vertical trace gas distribution is achieved by several transects at different heights but the same distance (Cambaliza, 2015; Karion et al., 2015; Pitt et al., 2019). Single point sources or small areas can be assessed by circular flight paths at different heights (Conley et al., 2017; Tadić et al., 2017; Ryoo et al., 2019). Using airborne eddy covariance technique, vertical fluxes can be inferred directly (Hiller et al., 2014; Yuan et al., 2015). Further techniques for airborne emission estimation include active and passive remote sensing instruments (Amediek et al., 2017; Krautwurst et al.,

2017). All methods can be combined with inverse modelling to derive emission distributions (Kort et al., 2008; Polson et al., 2011; Brioude et al., 2013; Xiang et al., 2013; Cui et al., 2015).

This study is part of the Carbon Dioxide and Methane (CoMet) mission. The goal of CoMet is to develop and evaluate methods for the independent monitoring of greenhouse gas emissions and to provide data for satellite validation. CoMet combined a suite of airborne active (lidar) and passive (spectrometers) remote sensors with in situ instruments to provide

local- to regional-scale data about atmospheric concentrations of $CO_2$ and $CH_4$ and to derive emissions on different spatial scales. One of the foci of CoMet was the Upper Silesian Coal Basin (USCB), located in southern Poland, which represents one of the largest European $CH_4$ emission sources with a total of around 500 kt $CH_4$/a (~3% of European $CH_4$ emissions), emitted from about 40 hard coal mines (EEA, 2019). $CH_4$ is released from the coal deposits and bedrock before and during mining and ventilated to the atmosphere through individual ventilation shafts due to safety reasons (Figure 1). The USCB is

also a heavily industrialized urban agglomeration of >2 million inhabitants. During the CoMet mission in early summer



2018, we performed airborne in situ measurements of $CH_4$, $CO_2$ and CO aboard the DLR aircraft Cessna Grand Caravan 208B using a well-established cavity ring-down spectrometer (CRDS) and a modified quantum and interband cascade laser spectrometer (QCLS, Kostinek et al., 2019).

During ten research flights conducted in May and June 2018, we studied emissions from coal mine ventilation shafts, power plants and other industrial facilities in the USCB region by using an airborne mass balance approach. Depending on the wind situation, different areas of the USCB region were targeted. To account for the lower part of the emission plume not accessible by aircraft, a number of vans equipped with mobile in situ measurement systems conducted ground-based measurements in a coordinated manner. Here we present trace gas observations from the two mass balance flights targeting the emissions of the entire USCB, one in the morning and one in the afternoon of the same day, June 6, 2018. Since the

morning is not an ideal time for the in situ mass balance method because of the growing convective planetary boundary layer, we consider the estimate from the afternoon flight to be more reliable. However, we describe the morning flight as well and consider its results as additional information. In Chapter 2 we present the observational data used in this study to derive emission estimates, a theoretical description of the mass balance method including the statistical interpolation method kriging together with the uncertainty analysis, and an overview of emission inventories available for the USCB. Chapter 3

contains the results of the mass balance flights. It includes a presentation of the meteorological situation, as well as the mass balance estimate and its uncertainties. Chapter 4 compares our mass balance emission estimate with current emission inventories. A conclusion is given in Chapter 5.

## 2 Data and methods

### 2.1 Observational data

During the CoMet 1.0 campaign several aircraft and ground based instruments were used to extensively sample greenhouse gas emissions of the USCB in early summer 2018. Here we present measurements taken aboard the DLR Cessna Grand Caravan 208B (Caravan). The Caravan was based in Katowice, Poland, from May 29 to June 13, 2018. Ten research flights were conducted in the USCB targeting different parts of the USCB. The flight paths were planned using a $CH_4$-plume forecast provided by the online-coupled, three times nested global and regional MECO(n) model (Nickl et al., 2019). For our

estimation of entire USCB emissions, we use airborne in situ observations from two flights on June 6, 2018, one in the morning (09:22 - 11:45 UTC) and one in the afternoon (13:01 - 15:28 UTC), in the following referred to as flights A and B, respectively. Figure 1 shows the flight track of flight B on a map with the $CH_4$ emission sources. Both flights were designed in a box pattern with an upwind leg in the northeast approximately in the middle of the PBL and the downwind wall in the southwest with flight transects at several heights. $CH_4$, $CO_2$, and CO enhancements were clearly observed in the downwind

wall. The flights were conducted in coordination with ground-based teams, which drove the instrumented vans below the upwind and downwind legs. Their tracks and sampled $CH_4$ mole fractions for the afternoon flight are shown in Figure 1. For



the emission estimation, we selected ground-based data according to closeness in time. Sampling times for flight and ground-based data are listed in Table S1.

Additionally, three Doppler wind lidar instruments Leosphere Windcube 200S were stationed at Rybnik, Wisła Mala and
Krzykawka to measure vertical profiles of wind speed, wind direction and turbulence parameters (Figure 1). Details on the CoMet lidar wind measurement setup and the planetary boundary layer height (PBLH) determination are given in Wildmann et al. (2020) and Luther et al. (2019).

A sophisticated suite of instruments aboard the Caravan gathered both, meteorological parameters and trace gas concentrations. A 5-hole probe, connected to a pressure transducer, is mounted on a nose boom under the left wing of the
aircraft and measured the three dimensional wind vectors. The temperature, pressure, and humidity sensors and the calibration of the wind measurement system are described in detail by Mallaun et al. (2015). A flight-ready CRDS analyzer (G1301-m, Picarro) was installed in the cabin of the aircraft. It measured $CH_4$, $CO_2$ and water vapor at a frequency of 0.5 Hz with cavity ring-down spectroscopy. Trace gas concentrations for water vapor were corrected according to Rella et al. (2013). The calibration and uncertainty assessment were conducted in analogy to Klausner et al. (2020), who used the same
instrument, aircraft, and calibration technique. Details specific to the CoMet set-up can be found in the Supplement (Table S2 and Text S1). CO is measured with a modified QCLS (Aerodyne) that also records $CO_2$, $CH_4$, ethane ($C_2H_6$), and nitrous oxide ($N_2O$) (Kostinek et al., 2019). Furthermore, a dry air sampler with 12 glass flasks (1 l) was installed aboard the Caravan, which were filled during the flight and later analyzed in the laboratory at Max-Planck Institute for Biogeochemistry for trace gas concentrations and isotopic signatures ($CH_4$, $CO_2$, CO, $N_2O$, $H_2$, $SF_6$, $\delta^{13}C$-$CO_2$, $\delta^{18}O$-$CO_2$, $\delta^{13}C$-$CH_4$, $\delta^2H$-
$CH_4$). However, in this study we focus only on the continuous in situ observations, while the results of ethane measurements and isotopic signatures will be published in a follow-up study.

Ground-based $CH_4$ data were recorded by three teams using vans equipped with different CRDS analyzers (Picarro G2201-i, AGH University and University of Heidelberg; G2301, Utrecht University). The group from the AGH University measured below the upwind leg and groups from University of Heidelberg and Utrecht University sampled below the downwind
tracks. For traceability between airborne and ground-based systems, an instrument intercomparison was conducted with the same four gas cylinders.



**Figure 1: Flight track for flight B, color-coded with in situ measured CH$_4$ mole fractions. The wind was blowing from the northeast over the USCB (as indicated by the white wind barbs) carrying emissions to the south-west. Airborne observations averaged over 20 s are displayed as circles and mobile ground observations averaged over 80 s below the upwind track and the downwind wall are marked as triangles. Red markers show the locations of active coal mine shafts from the CoMet v2 inventory.**

## 2.2 Mass balance method

We use a mass balance method to calculate emission estimates for the USCB from two flights conducted on June 6. This approach is subject to several assumptions. First, the wind speed, wind direction, emissions, and the PBLH should remain constant over the sampling time. Second, the trace gas plume has to be discernible from the atmospheric background. Third, there shouldn't be any entrainment/detrainment into the free troposphere and the lifetime of the species must be much longer



than transport and sampling times. Finally, the trace gas plume should be well-mixed between the lowest flight track and the ground. These criteria are most likely to be met in the early afternoon, when the PBL has reached its maximum height and

does not rise any further. The PBLH generally increases during the morning; hence afternoon flights are preferred over morning flights for mass balance studies. For our morning flight, we determine the temporal change of the PBLH during sampling to be 20% of its final height. We apply a correction to the observed trace gas enhancements to account for this change (see Sect. 3.2).

In our approach we calculate the mass flux of each trace gas ($CO_2$, $CH_4$, and CO) through a vertical surface along the

downwind flight tracks, here called "wall" (see Figure 1). The wall stretches from the ground to the top of the PBL. Since the downwind measurements, ground-based and airborne, were not taken exactly on this wall, as a first step, all data used in the calculation, are projected onto the closest point of the wall and then interpolated to fill the entire wall using the well-known kriging approach. The flux through the wall is defined by

$$F = \int_{z=\text{ground}}^{z=PBLH} \int_{x=S}^{x=N} \Delta c_{x,z}\, v_{x,z}\, dx\, dz, \tag{1}$$

where $\Delta c_{x,z}$ is the concentration enhancement of the trace gas above the background at each grid point, while $v_{x,z}$ describes the wind speed component at each grid point perpendicular to the wall. The integration area is defined by the ground, the PBLH, and the edges of the wall to the south $S$ and north $N$ (see bottom right panel of Figure 2). The PBLH is determined from the vertical gradient of potential temperature, measured during profile flight sections, and the times when the top of the PBL was crossed in the wall. During the afternoon flight the PBL top was crossed three times in the wall and from this

information the slanted boundary layer height could be well constrained.

The concentration enhancements $\Delta c$ are calculated from observed, interpolated mole fractions $m$ and the background mole fraction $m_0$ of the trace gases using linear temperature and pressure profiles deduced from the airborne measurements:

$$\Delta c = (m - m_0)\, M\, \frac{p}{R\, T}. \tag{2}$$

Here, $M$ is the gas molecular weight, $p$ the pressure, $R$ the universal gas constant, and $T$ the temperature in Kelvin.

To retrieve trace gas mole fractions $m$ and wind speed $v$ on the wall between the actual flight tracks, we use the kriging interpolation method with a stochastic Gaussian model. Kriging creates a grid of estimated values from data points with sparse spatial coverage and also gives standard errors for these values. We use a modified version of the EasyKrig software (© Dezhang Chu and Woods Hole Ocean Institution). For more details see Mays et al. (2009) and Pitt et al. (2019), who previously used this software in an aircraft mass balance study.

For $CH_4$, not only the mole fraction measured along the flight transects but also the data of the ground-based measurements is included in the kriging. Although $CO_2$ was also measured on the ground by the same instruments, the data cannot be used because it is heavily influenced by the surrounding car traffic. For ground-based $CO_2$, neither large scale enhancements nor background concentrations could be discerned. We chose the $CH_4$ observations along the ground track closest in time to the airborne measurements. The data is projected onto the downwind wall, averaged over 20 seconds and then interpolated

horizontally to regular distances before kriging. Airborne data is averaged over 10 second intervals in order to reach similar





spatial resolution to the ground-based data. Only data below the PBLH is included in the kriging process. We then closely followed the approach described in Pitt et al. (2019). The kriging output fields of $CH_4$, $CO_2$, CO mole fractions and perpendicular wind speed are given at a grid resolution of 0.1° in latitudinal direction and 20 m in the vertical.

### 2.2.1 Downwind and upwind background determination methods

For the mass balance approach, the background mole fraction $m_0$ of the trace gases needs to be determined. Here we compare two methods: (i) background estimated from the downwind wall's edges and (ii) background estimated from the upwind leg. The downwind background method assumes that the boundary layer height remains constant for the time of sampling within the wall, while the upwind method requires the boundary layer to stay at the same height for the whole flight time and ideally a quasi-Lagrangian sampling of the same air mass in the upwind and downwind transects. Thus, the

less strict criteria of the downwind background method are more likely to be met in real conditions and we will use this method in our best estimate and the upwind background as a sensitivity test.

In order to determine the downwind background mole fraction from the wall's edges, we evaluate the variability of the $CH_4$ observations within the PBL. The background is separated from the plume using the standard deviation within a 2 min interval for airborne and 10 min interval for ground-based data. Starting at the edges of the wall, the interval is moved

towards the center. We define the boundary between $CH_4$ atmospheric background and plume where the standard deviation surpasses 3.4 ppb $CH_4$. The average $CH_4$ background standard deviation is 2.9 ppb. The $CO_2$ background section is adopted from the $CH_4$ background, because the variability in the background is too high for this approach to be applicable. The CO background threshold for the 2 min interval is 4.5 ppb with an average background standard deviation of 3.5 ppb. We average all background mole fraction observations within the PBL to the south and north of the plume separately. The mean

of these two values is considered as the average background for the downwind method. Thus, we assume a linear spatial gradient in the trace gas background.

The second way of determining the atmospheric background mole fraction uses the observations within the boundary layer from the upwind flight transect, which was flown about 15 minutes before the downwind wall and is here used in a sensitivity study. Methodologically, we define a perpendicular inflow transect according to the prevalent wind direction, and

project the upwind measurements onto this line (Supplement Figure S1). After interpolation to regular distances, the average inflow mole fraction represents the upwind trace gas background. This approach has the advantage that sources upwind of the area of interest can be identified through potential enhancements in the upwind transect and are excluded from the emission estimate. On the other hand, the upwind background assumes that the same air masses are sampled in the up- and downwind, which is not true for our two flights, since the air masses needed approximately 3-4 hours to travel form the

upwind to the downwind measurement location, while the aircraft only needed 15 minutes. The maximum time separation between up- and downwind sampling is 1.5 h. Thus, our sampling is not strictly Lagrangian (i.e. air mass following) and changes in boundary layer background concentrations over time may affect the emission estimates using the upwind





background method. Another disadvantage of using upwind background concentrations with respect to $CO_2$ is the necessity to account for large scale ground fluxes like the biogenic uptake of $CO_2$, which is discussed in the next section.

### 235  2.2.2  Simulation of biogenic uptake of $CO_2$

We derive the influence of biogenic uptake of $CO_2$ from a combination of backward trajectories, calculated using the Stochastic Time-Inverted Lagrangian Transport (STILT, Lin et al., 2003) model, and biospheric fluxes from Vegetation Photosynthesis and Respiration Model (VPRM; Mahadevan et al., 2008). STILT was set up with receptors distributed along the flight track of the downwind wall and from each receptor, we then release 100 particles in the model. To drive the
trajectory simulations, we used output of ECMWF HRES short-term forecasting system (approx. 9 km x 9 km spatial resolution, 137 vertical levels), preprocessed to assure mass-conservation of the wind fields. The median locations of the particle ensemble then constitute the median trajectories (Figure S2). As we are only interested in the influence from our domain of interest, we truncate the median trajectories at their position closest to the upwind measurement. The time lag between upwind and downwind sampling is approximately one hour, thus, the biospheric VPRM contribution to the
downwind measurements is finally calculated using the footprint derived from the last hour of each trajectory, multiplied with the VPRM fluxes corresponding in time and location. We add this contribution to the downwind $CO_2$ observation and then use these values for the interpolation with kriging.

### 2.3  Error estimate

For an error estimate of the derived mass flux we consider the statistical error of the input data and the systematic error of
the method.

### 2.3.1  Statistical error

The statistical error of our approach is determined using error propagation in the flux equation (Equations 1-2). The uncertainty calculation of the concentration enhancement $u_{\Delta c}$, the flux density uncertainty $u_{Fd}$ and the final flux uncertainty $u_F$ are described by equations 3-5:

$$\Delta c = c - c_0 \quad \rightarrow \quad u_{\Delta c} = \sqrt{u_c{}^2 + u_{c_0}{}^2} \ ; \tag{3}$$

$$Fd = \Delta c * v \quad \rightarrow \quad u_{Fd} = \sqrt{\left(\frac{u_{\Delta c}}{\Delta c}\right)^2 + \left(\frac{u_v}{v}\right)^2} * Fd \ ; \tag{4}$$

$$F = \sum_i Fd_i * A \quad \rightarrow \quad u_F = \sqrt{\sum_i \left(u_{Fd_i}\right)^2} \ * A \ . \tag{5}$$

The first two uncertainties are calculated for each grid point of the wall surface; the final flux uncertainty $u_F$ is the combination of the single uncertainties. The trace gas uncertainty $u_c$ and wind speed uncertainty $u_v$ are a combination of
measurement and kriging uncertainties expressed as kriging standard error (KSE):


$$u_{c/v} = u_{\text{measurement}} + \text{KSE} = u_{\text{measurement}} + \sqrt{u_{\text{kriging}} \cdot var(\Delta c)} \tag{6}$$

The measurement uncertainty $u_{\text{measurement}}$ has been determined to 1.1 nmol mol$^{-1}$ (hereafter referred to as ppb) for $CH_4$, 0.15 µmol mol$^{-1}$ (hereafter referred to as ppm) for $CO_2$ (Table S2, Text S1), and 7 ppb for CO (Kostinek et al., 2019). The wind speed measurement uncertainty $u_v$ has been assessed to be 0.3 m/s for each of the horizontal components (Mallaun et

al., 2015). The uncertainty of the interpolation and extrapolation kriging method is output by EasyKrig as a gridded field of normalized variance values $u_{\text{kriging}}$. To retrieve the gridded KSE (see Figure S4), which is the equivalent to the standard deviation, we multiply the kriging error output $u_{\text{kriging}}$ by the variance of the kriging input dataset $\Delta c$ and then take the square root (Equation 6). The background mole fraction uncertainty $u_{c_0}$ is here defined as the standard deviation of all data points contributing to the background calculation (see Table 4). The uncertainty of the grid cell area $A$ is assumed to be zero.

**2.3.2  Systematic error**

We conducted several sensitivity tests in order to test the robustness of our mass balance method and to determine its systematic error. These sensitivity tests are described and discussed in Sect. 3.4. We assume all systematic errors to be independent and calculate the total absolute systematic error as the square root of the sum of squared individual differences from the best estimate, which treats the data as described above using a downwind trace gas background as described in

Sect. 2.3.

**2.4  Bottom-up emission inventories**

Several inventories of greenhouse gas and air pollutant emissions exist for the USCB. They vary in spatial and temporal resolution, as well as in the time for which they are available. Table 1 gives an overview of the six inventories we use in this study for comparison with top-down derived $CH_4$, $CO_2$, and CO emissions in the USCB region.




**Table 1: Overview of emission inventories used in this study. The year states the last year, for which data are available.**

| Inventory | Year | Resolution | Coverage | Gases |
|---|---|---|---|---|
| E-PRTR v16 (EEA, 2019) | 2017 | point | Europe | $CH_4$, $CO_2$, CO |
| CoMet v2 (internal inventory) | 2016 | point | Silesia, CZ Moravia | $CH_4$, $CO_2$ |
| Scarpelli CH4 (Scarpelli et al., 2020) | 2016 | 0.1° x 0.1° | Global | $CH_4$ (Oil, Gas, Coal) |
| CAMS-REG v3.1 (Granier et al., 2019) | 2016 | 0.1° x 0.05° | Europe | $CH_4$, $CO_2$, CO |
| EDGAR v5/v4.3.2 (Crippa et al., 2018; Janssens-Maenhout et al., 2019) | see right | 0.1° x 0.1° | Global | $CH_4$ (2015), $CO_2$ (2018), CO (2012) |
| GESAPU (Bun et al., 2019) | 2010 | 15`` x 15`` (~400 m) | Poland, Ukraine | $CH_4$, $CO_2$, CO |

The first point source inventory listed in Table 1Table 1 is the European Emission Release and Transfer Register (E-PRTR). It results from the Regulation (EC) No 166/2006 which implements the United Nations Economic Commission for Europe
(UNECE) PRTR Protocol under which industrial facilities have to report their emissions to air if they exceed a threshold of 100 t/a for $CH_4$, 100 kt/a for $CO_2$, and 500 t/a for CO. Annual data can be downloaded from the European Environmental Agency's website (EEA, 2019). More information on the E-PRTR is given via its website: https://prtr.eea.europa.eu/ (last accessed: 24 February 2020).

The CoMet v2 inventory is a point source inventory based on the E-PRTR 2016 emissions created by the CoMet team
especially for this campaign. It comprises anthropogenic sources of $CH_4$ and $CO_2$ in the USCB and its vicinity. The largest difference between the E-PRTR and the CoMet inventory is that E-PRTR considers each coal mine as one single point source, often located at the mining operator headquarters, whereas in the CoMet inventory individual ventilation shafts were visually geo-localized using Google Earth. Then, the emission value of each mine was evenly distributed between all ventilation shafts belonging to that mine. Active Czech coal mines in the Ostrava region did not report any $CH_4$ emissions to





E-PRTR but were assumed to emit the same amount of $CH_4$ per ton of extracted coal as Polish mines. We deduced a factor
of $11.8 \pm 5.2$ kg $CH_4$ per ton of extracted coal for the USCB mines listed in Table S3 and applied this value to the Czech
mines of Karvina, Karkov, CSM, and Paskov. The locations of the fourteen listed landfills and waste disposal sites were
checked against satellite imagery. Their $CH_4$ emission is assumed to be 3.3 kt/a, which is less than 1% of the total USCB
emissions.

Scarpelli et al. (2020) published the newest gridded emission inventory available for comparison within in this study. It only
contains $CH_4$ emissions from oil, natural gas and coal exploitation. But since these are the main sources (87% according to
CAMS) of $CH_4$ emissions in the USCB, values are comparable to the total of other inventories. Scarpelli et al. (2020) use the
national totals of emissions reported to the UNFCCC and distribute them according to the positions of relevant
infrastructure. Uncertainties of the emissions are based on the emission factor uncertainties from the Intergovernmental

Panel on Climate Change (IPCC) and are given as gridded information. Averaged over the USCB, the given relative error
standard deviation for $CH_4$ emissions is 60.9%.

The Copernicus Atmospheric Monitoring System (CAMS) regional emission inventory (CAMS-REG-GHG/AP; Granier et
al., 2019) is based on the TNO-MACC inventories (Kuenen et al., 2014). This inventory offers a resolution twice as high as
the Scarpelli and EDGAR inventories. The inventory was also constructed by using the reported emission national totals by

sector and spatially distributing them consistently across all countries by using proxy parameters.

The most widely used gridded emission inventory is probably the Emission Database for Global Atmospheric Research
(EDGAR, https://data.europa.eu/doi/10.2904/JRC_DATASET_EDGAR) global emission inventory. The most recent version
5.0 (https://edgar.jrc.ec.europa.eu/overview.php?v=50_GHG) includes emissions of the three major greenhouse gases $CO_2$,
$CH_4$ and $N_2O$. It is based on the previous EDGAR version 4.3.2 (Janssens-Maenhout et al., 2019). We use the CO emissions

from the air pollutant inventory (Crippa et al., 2018) from version 4.3.2. The most recent year of emission data is 2015 for
$CH_4$, 2018 for $CO_2$, and 2012 for CO. In EDGAR, annual country-specific emissions are derived from international activity
data and emission factors, which are then distributed in time and space using monthly shares and spatial proxy datasets. The
data includes uncertainty factors per species for three types of countries: OECD countries of 1990, countries with economies
in transition in 1990, and the remaining countries in development. European emissions from EDGAR in 2012 have standard

deviations of 16 % for $CH_4$, 2.5 % for $CO_2$ (Janssens-Maenhout et al., 2019), and 65 % for CO (Crippa et al., 2018).

The GESAPU inventory (Bun et al., 2019) has been created for Ukraine and Poland only for the reference year 2010.
Originally, it is a point, line, and area source inventory based on shapefiles. The advantage of this type of information is that
it has a very high resolution, but can also be gridded with any spatial resolution and orientation. The GESAPU inventory
comprises all sectors of anthropogenic emissions. Here we use a gridded version of the emissions with a resolution of 15 arc

seconds (approximately 296 m x 463 m for the region).

Figure 2 shows the spatial distribution of $CH_4$ emissions as given by the six inventories. Point sources from E-PRTR and
CoMet inventory are displayed as black markers while the background colors give the gridded inventory values. Although
the inventories generally agree on the locations of $CH_4$ emissions, there are several cases, where sources seem to be missing.



Regarding point sources, E-PRTR (top left) has fewer individual sources than the CoMet inventory (top right) due to the
separation in single ventilation shafts. Additional mines in the CoMet inventory include the four Czech mines and the four
ventilation shafts of the Brzeszcze mine around 19.15°E and 49.95°N. The gridded Scarpelli (top left) emission distribution
for CH$_4$ does not represent the point sources well. There are no emissions north of 50.2°N although several mines are located
in this northern area. Generally the CAMS (top right) emission maxima seem to represent the point source locations better
than the Scarpelli or EDGAR (bottom left) emission distribution, with the exception of the Czech mines, which are included
in Scarpelli and EDGAR, but not in CAMS. In the GESAPU inventory, the high CH$_4$ emissions associated with mining
activities were visualized by overlaying marker for sources above 1 kt/a on the gridded emission map. These are fewer high
emitting sources than in the E-PRTR inventory. This could be caused by consolidation and separation of mines between
2010 and 2017, the respective years for the data. Two flights (on June 6, 2018), which are shown as blue tracks in Figure 2,
were designed to capture the emissions of the region during north-easterly wind conditions.

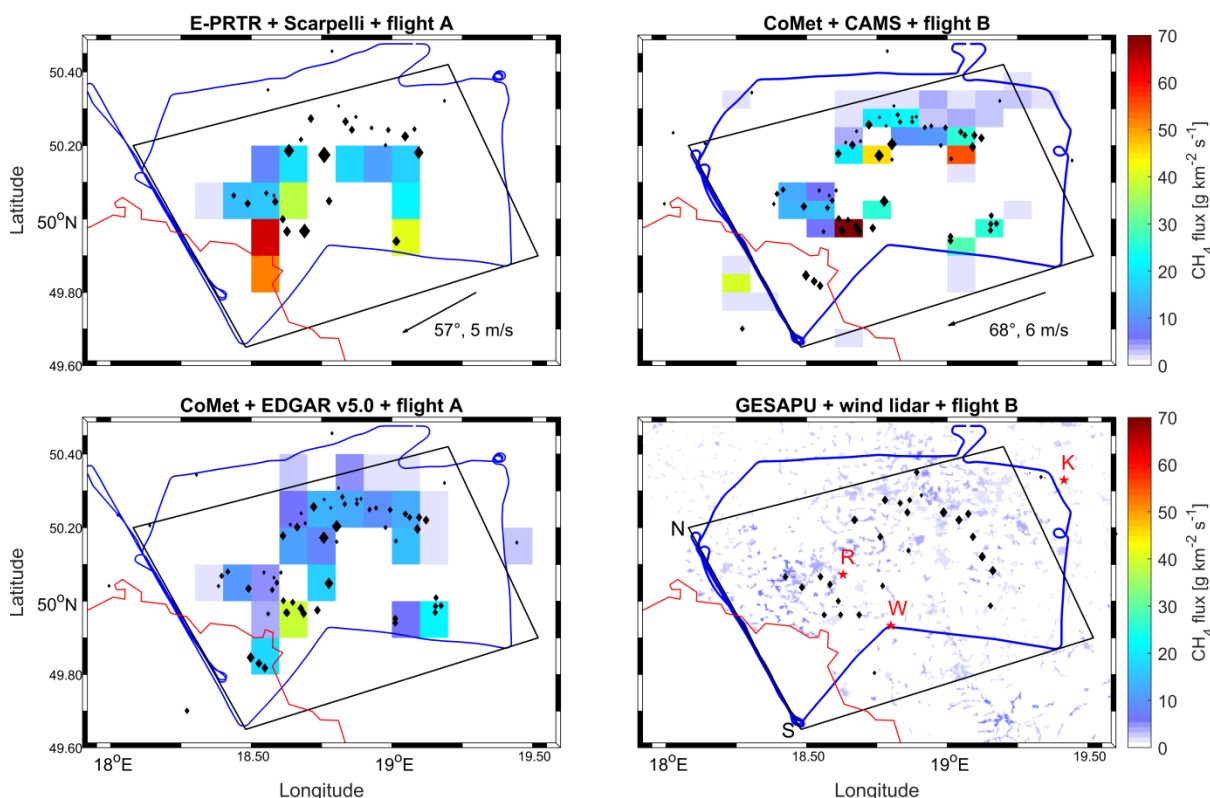

340

**Figure 2: CH$_4$ emission distribution of inventories in the USCB. Background colors give emissions from gridded inventories
Scarpelli, CAMS, EDGAR, and GESAPU, while the markers are sized according to the emissions of the point source inventories
E-PRTR and CoMet. Additionally, we added GESAPU sources above 1 kt/a CH$_4$ as markers for better visibility. The black boxes
denote the emission area for comparison with the mass balance estimate via aircraft. The blue lines show the flight tracks of the**
345    **flights A and B on June 6, 2018, used in the mass balance and the arrows in the top two panels show the mean wind direction
during the two flights. The red line denotes the Polish-Czech border. Red stars in the bottom right panel show the locations of the
wind lidar instruments (R: Rybnik, W: Wilsa Mala, K: Krzykawka). Also marked in this panel are the southern and northern
edges of the downwind wall S and N.**

The $CO_2$ and CO emission distribution in the inventories is displayed in Figure 3. $CO_2$ point sources (from E-PRTR and
CoMet) agree well with EDGAR and CAMS, except for the strong $CO_2$ and CO emissions associated with the Lagisza
power plant and Acelor Mittal steel factory at 50.34°N and 19.28°E, which are correctly placed in the northeast corner of the
flight track in E-PRTR, CoMet and GESAPU. Instead, EDGAR and CAMS include an emission hot spot to the southeast and
east, respectively, of this location, that is not associated with a point source. The Rybnik power plant, located in the central
western USCB, is the strongest point source emitter of $CO_2$ in all inventories. CO has one emission hot spot in the USCB,
namely the Acelor Mittal steel factory next to the Lagisza power plant with 137 kt/a in E-PRTR 2017. This source is not
represented in EDGAR and shifted to the east in CAMS. GESAPU includes this source, but with much lower emissions of
63 kt/a.

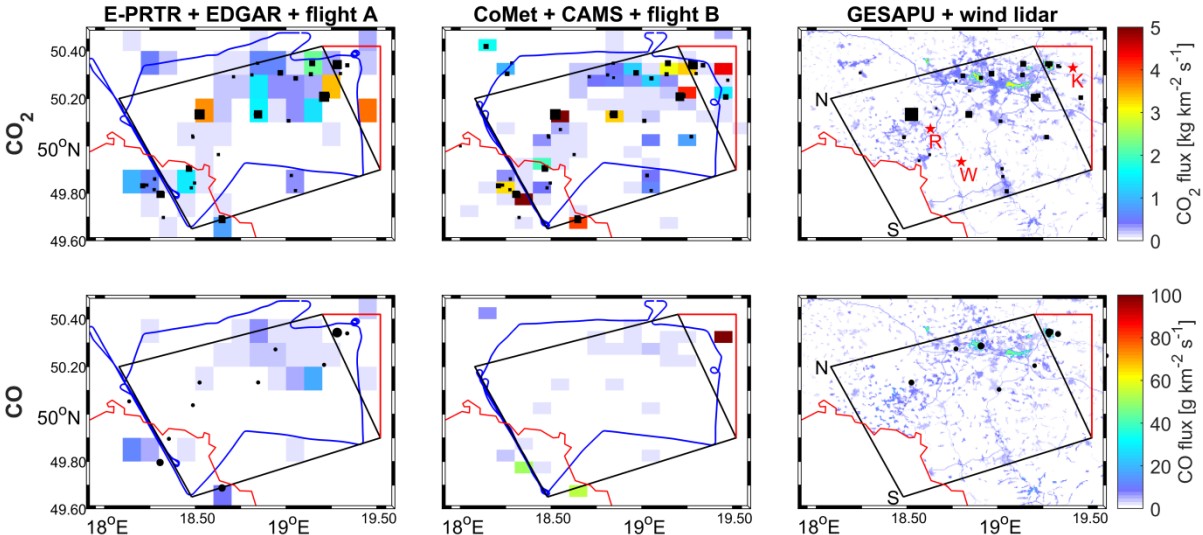

**Figure 3: Like Figure 2 but for $CO_2$ and CO. GESAPU sources above 0.1 Mt/a and 1 kt/a for $CO_2$ and CO, respectively, are added
as markers. The straight red lines show the addition to the mass balance area necessary because of misplaced sources.**

To compare the emission inventories with our mass balance flights, the emissions of each inventory are summed up within
an area representative of the flight track and wind direction (more details see Sect. 3.3), which is marked by the black boxes
in Figure 2 and 3. Since some of the $CO_2$ and CO sources are obviously misplaced in the gridded inventories, but really lie
within our mass balance area, we enlarged the mass balance area toward the east in order to include these sources into the
USCB sum. These enlargements are marked by red lines in Figure 3. Although missing sources influence the comparison
between inventories and the emission estimate via aircraft, the misplacements might not, since misplaced emissions are now
within the enlarged mass balance area.

For each inventory, the total annual emission from the enlarged area including the reported uncertainty is given in Table 2.
These values include emissions from all sectors available in the inventories (see also discussion in Sect. 4). Scarpelli
assumes the highest emissions for $CH_4$, followed by CAMS and CoMet. GESAPU features the lowest $CH_4$ emissions, which



might partly arise from the sources in the Czech Republic, which are not covered in the inventory. Highest $CO_2$ emissions are assumed by the EDGAR inventory. CO emissions are highest in CAMS, closely followed by GESAPU.

**Table 2: Annual emission totals in the USCB area for different emission inventories and trace gases.**

| Inventory | CH$_4$ [kt/a] | CO$_2$ [Mt/a] | CO [kt/a] |
|---|---|---|---|
| E-PRTR | 448 | 37.0 | 144 |
| CoMet | 581 | 39.1 | -- |
| Scarpelli | 685 ± 456 | -- | -- |
| CAMS | 621 | 51.5 | 329 |
| EDGAR | 556 ± 89 | 59.0 ± 1.5 | 236 ± 154 |
| GESAPU | 405 | 56.8 | 291 |

## 3    Results

### 3.1    Meteorological situation

The meteorological conditions have to fulfil certain criteria for a feasible mass balance calculation. On June 6, 2018, the weather conditions for an airborne mass balance experiment in the USCB were advantageous due to relatively constant wind speed and wind direction over the sampling time. The PBLH changed considerably during flight A in the morning, but was rather constant during flight B in the afternoon.

The wind lidar measurements at Rybnik airport were located close to the center of our in situ wall (Figure 2) and can be used to assess the wind history over the entire measurement day. Vertical profiles of wind speed and wind direction show that during the previous night a low-level jet blew over the area with wind speeds of more than 10 m/s, in the morning the wind slowed down to around 5 m/s and then accelerated to 6-7 m/s around 13:00 UTC (Figure 4, Table 3). The boundary layer wind direction was between 50° and 70° over the entire day. The nightly low-level jet prevented accumulation of emissions, and the slowing down around 6:00 UTC provided relatively constant wind speeds for four hours before we started our downwind sampling at 10:00 UTC. This steady wind history prior to the flight is crucial for the mass balance approach, because of the assumptions stated in Section 2.2. During this time emissions from the farthest shafts (75 km from downwind wall) were able to travel from emission to observation location at constant wind speed and direction. A comparison of aircraft observations in the downwind wall and wind lidar averages during the observation times is given in Table 3. Observed wind speeds with the lidar are within the range of aircraft observed wind speeds. Generally wind speeds in the southern USCB were about 1 m/s higher than in the northern part of the USCB.





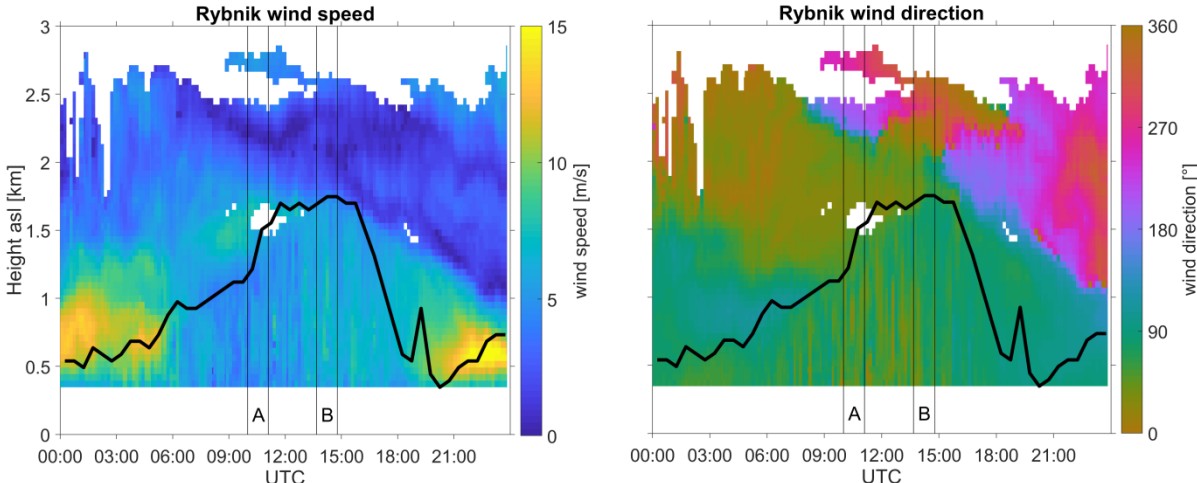

**Figure 4: Wind speed and direction at Rybnik measured with a Doppler wind lidar on June 6, 2018. The bold line denotes the**
**PBLH determined from the eddy dissipation rate and the thin vertical lines illustrate the downwind wall sampling times of flights**
**A and B.**

**Table 3: Overview of wind data and PBLH from aircraft averaged within the downwind wall and wind lidar observations at**
**Rybnik. Aircraft data give uncertainty ranges due to measurement uncertainty and wind lidar data state a standard deviation of**
**the measurements in the PBL.**

|  | Mean wind speed perpendicular [m/s] | | Wind dir. [°] | | PBLH [km asl] | |
|---|---|---|---|---|---|---|
|  | Aircraft | Wind lidar | Aircraft | Wind lidar | Aircraft | Wind lidar |
| Flight A (morning) | $4.8 \pm 0.3$ to $5.7 \pm 0.3$ | $5.0 \pm 0.9$ | $48 \pm 2$ | $57 \pm 15$ | $0.9 \pm 0.05$ and $1.25 \pm 0.05$ | $1.2 \pm 0.05$ to $1.5 \pm 0.05$ |
| Flight B (afternoon) | $5.8 \pm 0.3$ to $7.0 \pm 0.3$ | $6.4 \pm 0.8$ | $62 \pm 2$ | $68 \pm 12$ | $1.3 \pm 0.05$ and $1.8 \pm 0.05$ | $1.7 \pm 0.05$ |

The diurnal development of the PBLH, with a maximum of 1.7 km above sea level (asl), is discernible from the wind lidar

observations. The PBLH measured by the wind lidar increased from 1.1 to 1.5 km during the sampling of flight A, but

remained relatively constant at 1.7 km during flight B. We also determined the PBLH from two vertical aircraft profiles of

potential temperature, observed before and after the sampling of the downwind wall (Figure S3). Before flying the wall

pattern, we obtained a vertical profile in the southern part of the USCB area (around 18.2°E, 49.8°N). After finishing the

wall pattern a northern profile was sampled on the way back to Katowice airport (18.2°E, 50.3°N). During both flights, the

PBLH was about 400 m lower in the southern part than in the northern part of the USCB. Thus, the PBLH data in Table 3

describes a latitudinal gradient for the aircraft, and temporal changes from the wind lidar.





Furthermore, for the mass balance, we assumed no entrainment from the free troposphere during sampling time. This assumption is supported by a strong capping inversion at the PBLH observed in the aircraft profiles (Figure S3). Still, since the PBLH was increasing during the sampling for flight A, there was considerable entrainment of free tropospheric air into the mixed layer. The correction we applied for this temporal change of the PBLH is described in the following section. The uncertainty related to this correction is assessed in the sensitivity test (Sect. 3.4) concerning the temporal PBLH variability.

**3.2 Kriging results**

For our mass balance, we use airborne in situ observations from two flights on June 6, 2018. $CH_4$, $CO_2$, and CO enhancements were clearly observed in the downwind wall. The ground-based teams drove below the upwind and downwind legs using the closest highways and national roads. Halfway through the southern track we ascended and descended to derive the height of the PBL based on meteorological measurements. Above the PBL, observed $CH_4$ and CO concentrations were
lower than within the PBL, while $CO_2$ concentrations were higher.

In a first step of emission estimation for the entire USCB (as described in Sect. 2.2) the observed data in the downwind wall is inter- and extrapolated using the kriging algorithm (Figure 5). Details of the kriging parameters can be found in the supplement (Text S2). Mole fractions in the wall are cut off below the ground, above the PBL, and to the south and north of the flight legs (points S and N).

For the morning flight A, the trace gas plumes reach from the ground to the top of the PBL. The transects on the ground and at 800 m show the highest $CH_4$ maxima (Figure S4). At 1000 m and 1100 m the maximum enhancements are lower. The same is true for the $CO_2$ and CO enhancements. This is probably caused by the growing PBLH during the flight. During the downwind measurement of the morning flight A, the height of the PBL increased from 1.2 km asl (0.9 km above ground level (agl)) to 1.5 km asl (1.2 km agl), which is an increase of 20%. The lowest transect (800 m) was sampled first in the
shallowest PBL. The two upper transects were sampled about half an hour later, when the PBLH had increased by about 20%. Thus the emissions from the USCB were mixed within a much smaller volume during the lowest transect, than during the following two. The ground-based sampling of the morning flight took place between 9:00 and 10:40 UTC. Two cars started in the center of the downward projected flight track and moved away from each other to the south and north. Thus, the central part was sampled first, during low PBLH conditions. To account for the low PBLH during the first flight transect
and the ground-based sampling, we apply a correction factor of -20% to the ground observations and the lowest flight transect. Figure S4 shows the original, uncorrected observational data, while Figure 5 shows the corrected values. Corrected enhancements are in the order of 0.16 ppm $CH_4$, 7 ppm $CO_2$ and 130 ppb CO.

During the afternoon flight B, the $CH_4$ plume is evenly distributed between the ground observations and the lowest flight track at 800 m (Figure 6). Thus, we assume good vertical mixing within the PBL and use the same $CO_2$ and CO mole
fractions at the ground as in the lowest flight transect. Trace gas enhancements are in the order of 0.12 ppm $CH_4$, 6 ppm $CO_2$





and 120 ppb CO, thus, lower than during the morning flight. The main $CH_4$ plume is located at 50.0°N with a secondary plume around 49.8°N. There are two $CO_2$ plumes at 50.0°N and 50.1°N. The CO plume is located at 50.0°N.

The horizontal wind speed shows a latitudinal gradient with higher wind speeds in the south than in the north for both flights. This gradient is preserved when using a kriged wind field for flux calculation instead of an average wind speed for the whole downwind wall (as discussed in Sect. 3.4).


Error estimates from the interpolation and extrapolation are retrieved from the kriging software as gridded fields (see Figure S5). The KSE generally increases with distance to the measurement locations and is highest at the ground for $CO_2$, CO and wind speed because no ground-based measurements were available for these parameters.





**Figure 5: Mole fractions and perpendicular wind speed in the downwind in situ wall from observations (circles) and inter- and extrapolation with a kriging algorithm (shading). The CH₄ wall incorporates ground-based measurements. For CO₂ and CO the ground mole fraction is assumed to be the same as in the lowest flight track. The wind extrapolation does not use any information below the lowest flight track.**


### 3.3 Background mole fractions

We applied both the downwind and the upwind method (see Sect. 2.2.1) to determine atmospheric background mole fractions of trace gases. Average background mole fractions and standard deviations for both methods are summarized in Table 4. Figure 6 shows the observed PBL mole fractions of $CH_4$, $CO_2$, and CO at different heights for flight B. The highest transect (light blue), originally planned in the free troposphere above the PBL, turned out to partially be within the PBL, but the southern and northern end were sampled in the free troposphere. The background mole fractions according to the

downwind method are displayed as dotted lines. For flight A, the background could not be reached to the south of the downwind wall and only background values from the north were used for $CH_4$ and $CO_2$ (Figure S4).

**Table 4: Average background mole fractions and their standard deviations calculated with the downwind and upwind methods.**

|  | Downwind background | | | Upwind background | | |
|---|---|---|---|---|---|---|
|  | $CH_4$ [ppm] | $CO_2$ [ppm] | CO [ppb] | $CH_4$ [ppm] | $CO_2$ [ppm] | CO [ppb] |
| Flight A | $1.941 \pm 0.005$ | $402.7 \pm 0.9$ | $82.5 \pm 8.9$ | $1.944 \pm 0.006$ | $404.6 \pm 1.0$ | $81.6 \pm 8.5$ |
| Flight B | $1.944 \pm 0.007$ | $401.8 \pm 0.7$ | $110.5 \pm 5.2$ | $1.936 \pm 0.004$ | $402.8 \pm 1.8$ | - |

The upwind mole fractions (black lines) were shifted to the corresponding latitudes of the downwind wall based on the wind direction. The $CH_4$ upwind mole fractions follow the same north-south gradient as the downwind background (Figure 6, top). Around 49.94°N the $CH_4$ mole fraction is slightly enhanced in the upwind. There is a similar enhancement around 50.13°N in flight A (Figure S4). Due to the projection, these would be between 50.2°N and 50.3°N on the inflow track. The only source upwind of the inflow track in the inventories is the Trzebinia mine and power plant at 19.44°E and 50.16°N. We use

the ground-based observations below the upwind track (grey line) to confirm our aircraft observations. They show similar absolute values and a similar north-south trend to the airborne track. Additionally, there are three spikes between 49.73° and 49.78°N. These locations correspond to an inflow latitude of around 50.0°N and probably originate from sources close by, since they don't appear in the airborne observations. The largest peak most likely originates from the coal processing and waste water treatment facilities right upwind to the measurement route at 50.027°N and 19.438°E.

The $CO_2$ upwind background is higher than downwind mole fractions at both ends of the measurement transects but lower in the center, where the downwind plume was observed. The average upwind background of $CO_2$ is 2 ppm and 1 ppm higher than the downwind background for flight A and B, respectively. This discrepancy is caused by the biogenic uptake of $CO_2$ between the upwind and downwind transects. The impact of the biogenic sink is discussed below.

Upwind CO observations during flight B do not cover the complete transect due to a start delay of the QCLS. Thus, we did

not use the CO upwind background for this flight. The CO upwind observations for flight A show small variations resulting



in a background standard deviation of about 9 ppb. Here, the upwind CO measurements are smaller than downwind background values.



**Figure 6: Mole fractions of CH₄, CO₂, and CO at different heights above mean sea level within the PBL downwind of the sources for flight B. Background mole fractions according to the downwind method are displayed as dashed part of the lines at the edges. Additionally, the background according to the upwind method is shown in black and grey. Upwind data has been shifted to the respective downwind latitude. The CO upwind background stops at 50° N due to an instrument start up delay on this part of the track.**

The upwind background method calls for an estimate of the biogenic uptake of $CO_2$. We estimate this uptake from the STILT trajectories and the VPRM model (see Sect. 2.3.2). Figure S2 exemplary shows the truncated trajectories for the





800 m altitude transect of flight B. Trajectories for other transects and flight A are very similar. The biogenic uptake for each trajectory is determined from the last hour of transport. By subtracting the VPRM uptake from the corresponding downwind measurement (as the uptake is negative), one can obtain a downwind $CO_2$ concentration without biospheric influence. This uptake is on average 1.00 ppm for flight A and 0.95 ppm for flight B.

**3.4 USCB emission estimate**

From the two mass balance flights on June 6, 2018, we determined the total USCB emissions of $CH_4$, $CO_2$, and CO. Figure 7 summarizes the best-estimate emissions and the sensitivity calculations (see Sect. 2.3). The uncertainty of the best-estimate includes the statistical error, calculated from the uncertainties of the input parameters and the systematic error calculated from the sensitivity tests. The $CH_4$ emission estimates for the entire USCB on June 6 are $13.8 \pm 3.6$ kg/s and $15.1 \pm 3.0$ kg/s

for flights A and B, respectively. This is a difference of 9% between the two flights. The $CO_2$ emission estimates are $1.21 \pm 0.72$ t/s and $1.12 \pm 0.37$ t/s for the two flights, also with a difference of 9%, but with the morning flight results being higher. Finally, CO emissions from the USCB were calculated to be $10.1 \pm 3.2$ kg/s and $10.7 \pm 2.9$ kg/s for flight A and B, respectively. The discrepancy between them is 6%.







**Figure 7: USCB emission estimates on June 6, 2018, using an airborne mass balance approach including several sensitivity tests.**

We determined the systematic errors with several sensitivity tests applied to the treatment of different variables during the mass balance calculation (Figure 7). Systematic errors are calculated as emission difference between the best estimate mass balance using downwind background as described in Sect. 2.3 and the sensitivity studies:

**1) Upwind background method**

This background method leads to almost the same $CH_4$ emission estimate for flight A. The flight B estimate is 18% larger than the best estimate, showing that the assumption of a linear background gradient is not true for this case. The $CO_2$ emission estimate using an upwind background is 50% and 16% smaller than the best estimate for flights A and B, respectively. Especially for flight A, the upwind $CO_2$ mole fractions in the PBL might be enhanced due to a shallower PBLH. Also, the experiment was not conducted in a Lagrangian way, meaning that the sampling time difference between upwind and downwind does not match the travel time of the air. With potentially inhomogeneous biosphere-atmosphere





fluxes this could cause a problem. For CO the upwind background method yields an emission estimate difference of 3% for flight A. For flight B we did not calculate a CO emission estimate because of an incomplete upwind measurement (Figure 6). In general, CO upwind and downwind background data is quite similar.

**2) Average wind speed**

The impact of wind measurement treatment on the estimated mass fluxes was tested by using the averaged observed wind speed instead of the kriged wind field. This technique could for example be employed if no wind measurements were available and average model winds need to be used. The emission estimates for the morning flight are up to 4% lower and for the afternoon flight up to 13% higher than for the best estimate. Here the systematic change in the emission estimates is caused by the location of the plume in the wind field. During flight A, the plumes were located where the wind speed was

slightly higher than average (see Figure 5). Using the average wind speed, thus, results in a reduction of the emission estimates. During flight B, the plume locations were in a slow wind region with higher wind speeds to the south, especially for the $CO_2$ and CO plume. Using averaged wind speed, thus, enhanced the emission estimate. We highlight the importance of measuring the wind speed simultaneously with the mole fractions and using this spatial knowledge in the flux calculation.

**3) Ground data uncertainty**

Since we did not use $CO_2$ and CO from the mobile ground measurements, we calculated the sensitivity of our approach to the precise knowledge of ground-based data for $CO_2$ and CO. Assuming a 10% uncertainty of the ground value enhancements and increasing the kriging input ground values by this factor, results in a systematic error of 15-20%. This shows that a good approximation, or even better a measurement, of mole fractions below the lowest flight track is important for exact emission estimates.

**4) PBLH uncertainty**

Another sensitivity of our method is related to the knowledge of the PBLH and its variability. Its exact determination in the downwind wall is only possible when we cross its top during ascents or descents. This occurred once during the morning flight and three times in the afternoon. The PBLH is further constrained by vertical profiles before and after sampling the downwind wall and through the wind lidar observations. This data hints at temporal and spatial variations in the PBLH (see

Sect. 3.1). Based on this data we assign an uncertainty estimate of 100 m to PBLH. We account for the spatial PBLH uncertainty in the emission estimate by using a boundary layer 100 m higher than our best estimate. This is realized through cutting off the flux density field at this increased boundary. For this sensitivity test, discrepancies are between 5% and 12% for all three gases.

**5) Temporal PBLH variability**

The last sensitivity test accounts for the temporal variation of the PBLH during the morning flight A. The PBLH showed a temporal variability of 300 m, quantifiable from wind lidar measurements. We assess the uncertainty caused by the





temporally increasing PBLH for the morning flight by omitting the trace gas enhancement correction described in Sect. 3.2. The systematic error for flight A is between 21% and 23%.

On average, the uncertainty of the background mole fraction (up to 50%) and the uncertainty of mole fractions at the ground (15-20%) have the highest impact on the systematic uncertainty. For flight A, the changing PBLH introduces an additional 21-23% uncertainty to the emission estimates. Assuming that the single systematic uncertainties are independent of each other, the total systematic error of the emission estimate is calculated as the square root of the sum of squared individual uncertainties and is added to the statistical uncertainty. The statistical error is 1% for $CH_4$ and around 3% for $CO_2$ and CO

and, thus, small compared to the systematic errors of this approach. It is added to the systematic error to obtain the total error of the emission estimates. The $CH_4$ emission estimate has a total relative error of 26% and 21%, the $CO_2$ estimate of 60% and 33% and the CO estimate of 32% and 27% for flights A and B, respectively. The errors are always larger for flight A than for flight B, since the afternoon flight is more suitable for a mass balance experiment due to the temporally constant PBLH.

## 4   Comparison with bottom-up inventories

Hereafter we compare our airborne top-down emission estimate for the USCB with the bottom-up emission inventories described in Sect. 2.1. Both emission values, the bottom-up inventory and the top-down mass balance estimate, are based on different methods and assumptions which hamper a one-by-one comparison. Especially differences in the temporal resolution of the two methods create a problem in case emissions are subject to strong temporal fluctuations such as a

seasonal or diurnal cycle. Aircraft-borne top-down methods can only provide snapshot emission estimates, which for a comparison need to be scaled to the temporal resolution of the emission inventories. At the same time, bottom-up inventories also include uncertainties, for example in the emission factors which are often derived from process studies and are then used to derive annual sums. For this comparison, we scale our mass balance emission estimate, based on a snapshot of one day in the early summer, to an annual emission estimate. We assume this scaling to be representative, because of the nature of the

USCB emissions. In general, coal mining activities continue all year round and the power plants using the excavated coal are continually operated base load facilities. Still, it is known that $CH_4$ emissions from individual ventilation shafts vary on weekly to monthly scale, when mines open new longwall excavation areas and ventilation increases. However, since we study emissions on a regional scale (including ~35 mines), we argue that emissions from individual shafts vary independently and therefore variations cancel out to a large extent. According to the CAMS inventory (Figure S6), industrial

emission, including coal mine exhaust, make up 87% of USCB $CH_4$ emissions, with the waste sector (11%) and fugitives (2%) being the other contributors. Thus, we assume our $CH_4$ emission estimate to be largely representative for the entire year. $CO_2$ emissions attributed to public power generation (65%) and residential heating (6%) do have an annual cycle. The other contributions include industry (21%) and transportation (7%). The CO emissions result to 30% from residential combustion with annual cycle with the remainder from public power (3%), industry (54%), and road transport (13%) without



annual cycle. Thus, there is an annual cycle for $CO_2$ and CO and our summer measurements likely underestimate the annual value. Additionally, gridded inventories need to be treated with caution when used in region-specific studies (Janssens-Maenhout et al., 2019). These inventories distribute national emission totals onto a grid using proxy data. Most of the uncertainty of the grid cell level data originates from the uncertainty in the proxy data (Hogue et al., 2016). Furthermore, the comparison of inventories from 2010 with observational estimates from 2018 is not consistent and we treat comparisons to
the GESAPU inventory with caution.

Our airborne mass balance $CH_4$ emission estimate on June 6, 2018, of $436 \pm 115$ kt/a and $477 \pm 101$ kt/a for flights A and B, respectively, is in the lower range of inventory emissions (Figure 8). E-PRTR emission estimates are similar to our estimate, despite the omitted sources with emissions lower than the threshold of 0.1 kt/a. The CoMet emission inventory is higher than both mass balance estimates, but within the error range of flight B. Compared to E-PRTR from 2017, the CoMet inventory
includes several mines in Poland that reported higher $CH_4$ emissions in 2016 than in 2017, three additional Czech mines, and four landfills within the mass balance area. Scarpelli, CAMS and EDGAR $CH_4$ estimates are also higher than our mass balance results. The GESAPU inventory states the lowest emissions, which may result from the missing emissions from Czech mines (estimated to be around 70 kt/a).

Our $CO_2$ aircraft mass balance emission estimates of $38.3 \pm 22.8$ Mt/a and $35.2 \pm 11.7$ Mt/a agree with all inventories within
the reported errors of the measurements. These errors are large, especially for the morning flight. Under very good conditions it is possible to report results that can inform about the quality of emission inventories, but issues like the biospheric fluxes of $CO_2$ and annual cycles of emissions impede comparisons to annual emission inventory values.

The CO emission estimates of $317 \pm 100$ kt/a and $339 \pm 90$ kt/a from the aircraft mass balance on June 6, 2018, are at the upper end of the emission inventories. Especially the E-PRTR emission estimate for 2017 is much lower than the mass
balance result. This point source inventory does not include emissions from the transport and residential sector, which together comprise 42% of USCB CO emissions according to CAMS (Figure S6), which explains the discrepancy. CAMS, EDGAR, and GESAPU inventories are in the range of the emission estimates, but due to the annual cycle in residential combustion we suspect that these inventories underestimate CO emissions from the USCB.





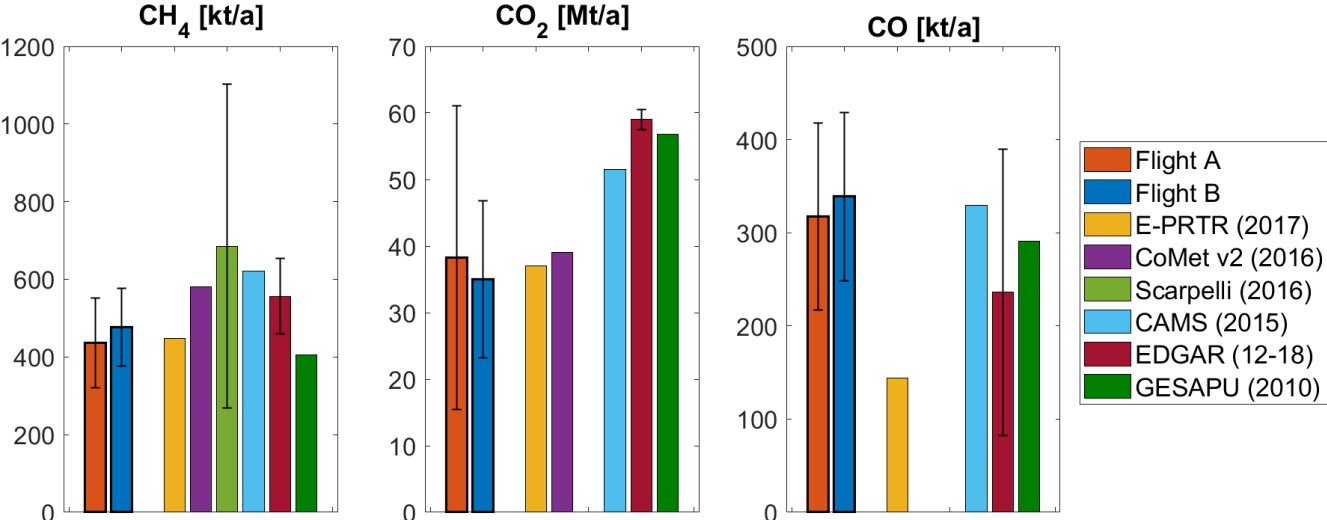

**Figure 8: Comparison of USCB emission estimates of the CoMet mass balance flights A and B with bottom-up emission inventories. Error bars show one standard deviation of the estimates, where available.**

## 5  Summary and Conclusions

In times of rising atmospheric concentrations of greenhouse gases and countries trying to reduce their associated emissions, it is important to develop an independent and objective emission monitoring system. During the CoMet campaign the European $CH_4$ emission hot spot Upper Silesian Coal Basin (USCB) was sampled by in situ techniques as well as passive and active remote sensing on ground and from aircraft. From two flights A and B around the USCB, conducted on June 6, 2018, combined with vehicle-based ground measurements, we determined a regional emission estimate of $CH_4$, $CO_2$, and CO for the entire USCB using in situ data and a mass balance approach. The plumes of all three trace gases could be observed and separated from the atmospheric background in all downwind transects. For the morning flight A, a trace gas enhancement correction was employed to account for the temporal change of PBLH during the sampling. We employed a kriging algorithm for the interpolation of observed $CH_4$, $CO_2$, CO and wind speed between the flight transects and towards the ground. $CH_4$ ground-based observations confirmed the existence of a well-mixed PBL with similar trace gas enhancements at the ground and in the aircraft transects. From the kriged fields we calculated the USCB emission estimate as the mass flux through the downwind wall for each flight. Using error propagation and several sensitivity tests we carefully determined the total error of our mass-balance approach. The $CH_4$ emission estimate has a total relative error of 21-26%, the $CO_2$ estimate of 33-60% and the CO estimate of 32-37%. These uncertainties are mainly caused by the background determination and the missing knowledge of mole fractions below the lowest flight track for $CO_2$ and CO. The higher uncertainty values apply to the morning flight estimate, because the temporal variation of the PBLH introduced a large error. Thus, we highlight the importance of a constant PBLH over time, knowledge of trace gas mole fractions at the ground and



the exact knowledge of background mole fractions. The large uncertainties in the $CO_2$ estimate are dominated by the uncertainties in biospheric $CO_2$ fluxes. These estimates could be improved by performing flights in wintertime, when the biospheric fluxes are negligible. Flights during different seasons would also better constrain the annual cycle in $CO_2$ emissions from the residential sector.

The CoMet in situ $CH_4$ emissions estimates from June 6, 2018, of $13.8 \pm 3.6$ kg/s and $15.1 \pm 3.0$ kg/s for flight A and B, respectively, are in the lower range of the six presented emission inventories. This agreement of our independent USCB emission estimate with the bottom-up coal mining emission reports indicates that this sector of emissions is well understood and monitored on regional scales. The emissions of $CO_2$ were determined to be $1.21 \pm 0.72$ t/s and $1.12 \pm 0.37$ t/s. The estimate from the second flight constrains the emissions to the lower end of inventory values. The gridded inventories, which

report higher emissions than our estimate, do not include an annual cycle in the residential combustion emissions of $CO_2$. This might be reflected in our low summer emission estimate. In general, an airborne mass balance estimate for $CO_2$ on these spatial scales is difficult due to inhomogeneous biospheric uptake. CO mass balance emissions of $10.1 \pm 3.2$ kg/s and $10.7 \pm 2.9$ kg/s for the USCB on June 6, 2018, are much higher than the E-PRTR point source inventory, which does not include residential combustion and road transport emissions, and are still in the upper range of the gridded emission inventory

values. The comparison between the snapshot top-down emission estimate and annual bottom-up inventories is influenced by the temporal variability of emissions in the USCB. Therefore, additional measurements during different seasons are needed to finally confirm bottom-up emission inventories.

Our airborne in situ mass balance method describes a measurement and evaluation strategy, which can be applied for various emission sources on a local to regional scale. In this case, we provide an independent bottom-up emission assessment for the

USCB, which also serves as a point of reference for other state of the art techniques, like airborne lidar and passive spectroscopy. A comparison of in situ and remote sensing emission estimation techniques will follow in future studies.

Independent top-down validation of emissions in industrialized countries can confirm the statistical approaches used in bottom-up inventories. Once facility locations and activity, technology and abatement information becomes available for other countries or regions, the confirmed emissions from industrialized areas will help to improve global emission

inventories used in climate projections. These will in turn help policy makers to develop efficient climate mitigation strategies. Consistent, reliable, and timely information on greenhouse gas emissions will allow the implementation, evaluation, and management of long-term policies that might allow keeping the global temperature rise below 2°C above pre-industrial levels.





## Author contributions

Alina Fiehn, Julian Kostinek, and Maximilian Eckl performed the trace gas measurements, calibrations and data preparation. Alina Fiehn analyzed the data and drafted the manuscript. Theresa Klausner provided shaft-wise E-PRTR geolocation and emission data retrieved from the E-PRTR dataset and the Polish State Mining Authority for 2014. This dataset was updated and expanded by Michal Galkowski to the used version 2. Michal Galkowski, Jinxuan Chen, and Christoph Gerbig provided STILT and VPRM simulations and helpful discussions on biogenic uptake of $CO_2$. Thomas Röckmann coordinated the deployment of ground-based measurements and helped with data evaluation and interpretation. Hossein Maazallahi, Martina Schmidt, Piotr Korben, and Jaroslaw Necki conducted ground-based in situ measurements in the field during the campaign and collected and shared the data. Pawel Jagoda conducted ground-based observations and supported the aircraft observations through coordinating the airport communications at Rybnik airport. Norman Wildmann took an active part during the campaign deploying the wind lidar and retrieving and providing wind lidar data. Christian Mallaun supervised the wind measurements onboard the Cessna Caravan and prepared the data. Rostyslav Bun provided a gridded version of the GESAPU emission inventory for the USCB. Anna-Leah Nickl and Patrick Jöckel devised, set up, and supervised the forecasting system that allowed flight planning for the CoMet campaign in the USCB. Andreas Fix coordinated all CoMet campaign contributions. Anke Roiger developed the research idea and coordinated the CoMet Cessna campaign operations. All authors contributed to the interpretation of the results and the improvement of the manuscript.

**Competing interests:** The authors declare that they have no conflict of interest.

**Acknowledgement:** The authors especially thank DLR-FX for the campaign cooperation, especially the pilots Thomas van Marwick and Philipp Weber and the group of Ralph Helmes, Andreas Giez, Martin Zöger, and Martin Sedlmeir. We would like to thank Joseph Pitt for providing an updated version of the kriging package and giving advice on its usage. We acknowledge funding for the CoMet campaign by BMBF (German Federal Ministry of Education and Research) through AIRSPACE (FKZ grants no. 01LK1701A and 01LK1701C). We thank DLR VO-R for funding the young investigator research group "Greenhouse Gases". The ground-based measurements on vehicles were funded by the European Union's Horizon 2020 research and innovation program under the Marie Skłodowska-Curie ITN project Methane goes Mobile – Measurements and Modelling (MEMO[2]; https://h2020-memo2.eu/) grant agreement no. 722479. The authors acknowledge ECCAD for archiving and distributing the CAMS emission inventories.

**Data Accessibility:** The data can be inquired directly from the authors and will be made public with the final manuscript.



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
