# Peer review of "Estimating CH4, CO2, and CO emissions from coal mining and industrial activities in the Upper Silesian Coal Basin using an aircraft-based mass balance approach"

_Atmospheric Chemistry and Physics, 2020_

## Referee Comment (RC1) · Zachary Barkley (Referee) · 5 Jun 2020

Review of "*Estimating CH4, CO2, and CO emissions from coal mining and industrial activities in the Upper Silesian Coal Basin using an aircraft-based mass balance approach*"

The authors describe 2 mass balance flights performed during a single day over a coal basin in Poland. Despite only having data for a single day, the analysis done is extremely thorough for this type of study, providing an extra layer of confidence to the overall solution, and serve as a sanity check for bottom-up inventory estimates of $CH_4$ from coal in the region (and helpful guidance for other trace gases). Nothing about the results are particularly remarkable, but it's good, necessary science nonetheless, and well-written at that. In its current state, I have no objective with publishing this paper after some extremely minor revisions are addressed.

**Minor comments**

Line 110: "*Since the 110 morning is not an ideal time for the in situ mass balance method because of the growing convective planetary boundary layer, we consider the estimate from the afternoon flight to be more reliable. However, we describe the morning flight as well and consider its results as additional information.*".
The later, afternoon loop tends to be more reliable for a number of reasons (morning blobs everywhere!). When I read this statement, I was bracing myself for crazy signals that cast doubt on the entire study. But such a thorough job regarding capturing the signal and meteorology from the first flight (as well as the period before it) that I would argue you're underselling your loop 1 results with this comment. To me, loop 1 and loop 2 together provide pretty good confidence in your calculated total for the day. I'd consider dropping these sentences entirely.

Line 126: It would be helpful to mention here the local time relative to UTC time.

Line 207: The background downwind method also requires the assumption that there are no sources upwind of your area of interest that would create a complex concentration pattern flowing into your domain. With that said, you have an upwind here and it's pretty clean, so it's obviously not an issue here.

Line 385: I just wanted to say I appreciate you mentioning the overnight winds, because so many mass balances neglect possible accumulation from stagnant winds in the overnight/dawn hours, leading to massive enhancement blobs scattered throughout the observations.

**Minor comments bonus points:**
This study does a remarkable job ensuring the validity of the mass balance technique by performing multiple vertical transects and even driving underneath the flight path to capture the signal at the surface layer. Many mass balance studies do not put in this level of effort, and it

would be good to know how necessary these extra precautions are with regard to calculating the true emissions. Furthermore, if we're going to use mass balance techniques at any point to verify emissions from a policy enactment standpoint, we'd want to be as efficient as possible with resources. So what I'd be curious about is, if you took the central transect from each loop and calculated the emissions using the simple assumption of a perfectly mixed boundary layer, how different would your solution be compared to your kriging results? Such a simple comparison would be useful to have in your uncertainty analysis and increase the scientific impact of your findings.

**Grammar**
Line 43: change "affect" to "affecting"
Line 473: "don't" to "do not", because a bunch of grumpy scientists gathered together in a room and decided that the use of contractions makes our science less impactful.

---

## Referee Comment (RC2) · Anna Karion (Referee) · 20 Jun 2020

This is a well-written, well-researched manuscript and should be published in ACP after 2 major corrections noted below. Overall, the authors have done a lot of work on this flight data (although only 2 flights). But especially given that there are only two flights, there are two additional revisions that should be considered.

1) Characterization of the wind conditions prior to the flights. This is done qualitatively ("relatively steady"), but not quantitatively, and can be used in the sensitivity analysis and uncertainty calculation. The mass balance requires the wind to be constant (Eulerian equation). If it is not constant over the transit time, there is uncertainty.

2) Use of biosphere model in sensitivity analysis. This is less important to the final result, but in the sensitivity analysis using the upwind transect as the background condition, the method of use of the biosphere model is confusing. The authors only subtract the biospheric influence one hour prior to measurement, because the upwind measurement was made one hour prior. This needs more explanation to me at least. However, since this method is only used to look at the sensitivity of background choice, it is a more minor issue than the above.

More discussion on both these points is below, with other detailed/minor comments.

L23: "estimates... which are well within the range"...

L60: e.g. reads awkwardly, I would write " which are used in climate projections, for example.".

L87 should say "using an airborne eddy covariance" (or maybe "the airborne eddy covariance...").

Figure 1. Could there be an inset to show the larger map where this is located? The google earth image does not actually show any location boundaries etc, or lat/lon indicators.

L102: make and model or reference for "well-established" CRDS? Were all 3 gases measured by both instruments? [ I see now this is further described in Section 2, so perhaps just refer here to that section ].

L108 - this is a very nice aspect of this study that is usually not done!

L112: Perhaps Chapters should be "Sections"? (this is up to the editors). Later the text does refer to "Sections", so consistency would be good.

L124: Earlier several remote sensing instruments were mentioned, but this work only

focuses on the in-situ measurements. Is there a reason for this?

L138 I don't think there should be a comma after both.

L165-166: The wind speed (and direction) also must remain constant over the transport time. This is a key assumption that the flux measured out of the downwind plane (or "wall") is equal to the emissions flux from the surface. Variability from this assumption is likely to happen, so that should be accounted for in the uncertainty analysis. Using wind speeds measured by the aircraft in the afternoon may not be the correct approach if they are not representative of the wind field over the whole domain over the transit time. If downwind wind is used, it should be shown that the wind speed was constant (or what the variability was) through the time it took for the air to transit the domain. Fig 6(a) in Karion et al., ACP, 2019 (https://doi.org/10.5194/acp-19-2561-2019) shows (granted, in an extreme case with long residence time ) the difference between true emissions and measured emissions in a downwind plane using a forward model. Given the relatively small domain here, it's likely fine to assume steady winds over transport time, but this should be stated (and the variability of the wind included in uncertainty/sensitivity analysis).

L211, using the upwind as a "check" has been done before, citing some literature here would strengthen the justification of this choice.

L229 "form" should be "from"

Section 2.2.2. It is not clear to me why only the last hour of the footprint is used to estimate influence of biogenic fluxes on the downwind transect. For the final mass balance, the edges of the plume are used as the background, so it don't understand why the time between the upwind and downwind sampling matters. For the case of using the upwind transect, I am still not clear on the use of the 1 hour time frame. More on this in the next section.

L264: Very nice detailed error analysis: This is where I believe that uv should include

the variability of the wind over the transport time, not just measurement error. Perhaps added in quadrature to the 0.3 m/s, and constant over the whole grid. Or, now reading 2.3.2, perhaps it should be considered in the sensitivity portion, as it would be a systematic error.

L283: Table 1 repeated

L328: no comma after 'cases'

L388 - Agree! This addresses my issue above about steady winds over the transit time - so the section 2.2 above should mention this as well. In table 3, I would recommend also showing the mean wind and standard deviation for the Lidar over the 3-4 hours transit time, not only the flight time (to address my earlier comment). Several times this section says "relatively steady" - using the lidar data, this can be quantified and stated here.

L429-435. This is one issue with Kriging in space samples that were conducted at different times! It seems that hopefully the correction did the job, but in general, one might think this is a reason to do separate mass balance calculations for each transect (with each its own PBL depth), and then average. But either way changing conditions are difficult to deal with, and the authors have made a decent attempt at accounting for this, so no need to do this.

Figure 5 these are nice useful figures. What is the averaging time prior to Kriging? Or is it the native 0.5-Hz data that is Krigged?

L490. To clarify the method for using VPRM above, is it not accounted for at all in the version where the edges are used as background, and only when the upwind background is used? This makes sense if the assumption is that the uptake is the same at the edges and in the center, which in this case is a reasonable assumption (maybe not in an urban area). Does the enhancement from VPRM not change across the transect? That would then justify not including it when using the edges as background.

Regarding the upwind backround use of VPRM: I am still confused by the use of one hour only, and given how variable the uptake is in time due to the diurnal cycle of fluxes, it's not clear how to deal with this. If biogenic fluxes were constant in time and space, then the full trajectories (footprints) should be used until the location of the upwind leg. Otherwise, I think this is too complex to handle in the method it has been handled here. If the upwind leg were flown one hour *after* the downwind legs then how would this work? I am wondering if the upwind transect just cannot be used other than to show the lack of significant upwind fossil sources. Please explain/justify the use of the one hour time frame, and how this would work if the upwind leg were flown simultaneously or after the downwind leg - would you not subtract VPRM at all in that case? Seems wrong.

L515: Definitely agree with this statement on biosphere-atmosphere fluxes being complex!

L520 (sensitivity using average wind speed during downwind legs): This is close to what I suggested earlier. However, this is the average over the flight time, not transport time. One could use the transit time instead (from lidar perhaps, or from the model), as was done by Karion 2013 & 2015, to see the effect. What matters more is what the wind was at the location and time of emission - if it was high, then less CH4 was picked up in that air mass, if low, vice-versa.

SI: Table 1 can the caption also explain the offset of local time from UTC so the reader can quickly translate the UTC hours to local?

How frequent is "frequently calibrated" (approximately - daily? hourly?). Text S1 gives the mole fractions of the cylinders in ppm for both gases, but the text says ppm/ppb. Should be ppm/ppm.

The drift with time uncertainty is estimated using the flight time, so presumably the calibrations did not occur during flight, so frequently is > 2.5 hours?

---

## Author Comment (AC1) · 24 Aug 2020

**Answers to reviewers for "Estimating CH4, CO2, and CO emissions from coal mining and industrial activities in the Upper Silesian Coal Basin using an aircraft-based mass balance approach"**

We would like to thank the two reviewers for the suggestions to improve the manuscript. Below you find our answers to their comments. The reviewer's comments are written in normal font, our answers in italics.

**Review 1 by Zachary Barkley**

The authors describe 2 mass balance flights performed during a single day over a coal basin in Poland. Despite only having data for a single day, the analysis done is extremely thorough for this type of study, providing an extra layer of confidence to the overall solution, and serve as a sanity check for bottom-up inventory estimates of CH4 from coal in the region (and helpful guidance for other trace gases). Nothing about the results are particularly remarkable, but it's good, necessary science nonetheless, and well-written at that. In its current state, I have no objective with publishing this paper after some extremely minor revisions are addressed.

**Minor comments**

Line 110: "*Since the morning is not an ideal time for the in situ mass balance method because of the growing convective planetary boundary layer, we consider the estimate from the afternoon flight to be more reliable. However, we describe the morning flight as well and consider its results as additional information.*" The later, afternoon loop tends to be more reliable for a number of reasons (morning blobs everywhere!). When I read this statement, I was bracing myself for crazy signals that cast doubt on the entire study. But such a thorough job regarding capturing the signal and meteorology from the first flight (as well as the period before it) that I would argue you're underselling your loop 1 results with this comment. To me, loop 1 and loop 2 together provide pretty good confidence in your calculated total for the day. I'd consider dropping these sentences entirely.
*We have deleted these sentences.*

Line 126: It would be helpful to mention here the local time relative to UTC time. – *Done.*

Line 207: The background downwind method also requires the assumption that there are no sources upwind of your area of interest that would create a complex concentration pattern flowing into your domain. With that said, you have an upwind here and it's pretty clean, so it's obviously not an issue here.
*We added the following sentences: "The downwind method also requires that there are no sources upwind of the area of interest which would create a complex concentration pattern flowing into the domain. This is shown in our upwind flight transect."*

Line 385: I just wanted to say I appreciate you mentioning the overnight winds, because so many mass balances neglect possible accumulation from stagnant winds in the overnight/dawn hours, leading to massive enhancement blobs scattered throughout the observations.
*Thank you for this comment!*

**Minor comments bonus points:**
This study does a remarkable job ensuring the validity of the mass balance technique by performing multiple vertical transects and even driving underneath the flight path to capture the signal at the surface layer. Many mass balance studies do not put in this level of effort, and it would be good to know how necessary these extra precautions are with regard to calculating the true emissions. Furthermore, if we're going to use mass balance techniques at any point to verify emissions from a policy enactment standpoint, we'd want to be as efficient as possible with resources. So what I'd be curious about is, if you took the central transect from each loop and calculated the emissions using the simple assumption of a perfectly mixed boundary layer, low different would your solution be compared to your kriging results? Such a simple comparison would be useful to have in your uncertainty analysis and increase the scientific impact of your findings.
*We did this analysis and added it as Section 3.5.*

**3.5 Single transect emission estimates**

*"This detailed calculation of the emissions can help to understand uncertainties of a mass balance in cases where less information is available. We ensured the validity of the mass balance technique by performing multiple vertical transects and even driving underneath the flight path to capture the signal at the surface layer. Many mass balance studies do not put in this level of effort, but it would be good to know how necessary these extra precautions are with regard to calculating the true emissions. Furthermore, when using mass balance techniques at any point to verify emissions from a policy enactment standpoint, we need to be as efficient as possible with resources. So, using the single transects within the boundary layer from each flight we calculated the emissions under the simple assumption of a perfectly mixed boundary layer. The PBLH was kept constant for all transects. Figure 1 shows the results of the single transect mass balance calculations for the two flights on June 6, 2018. The average of the single transects (blue) is always well within the uncertainty range of the kriging mass balance results (red). Nevertheless, single transect emission estimates deviate up to 40% in both directions from the kriging estimate for $CH_4$. This deviation is much larger than the kriging estimate uncertainty. Deviations are largest for transects close to the PBLH when the concentration gradient between the boundary layer and free troposphere is also large, e.g. the highest $CH_4$ transects. Thus, when calculating emissions from single transects the flight altitude should be well below the PBLH to avoid sampling free tropospheric air masses. On the other hand, these results discourage single transect mass balance estimates anyway."*

[Figure]

*Figure 1: Mass balance results for single transects compared to the average of all single transects and the kriging mass balance result from Section 3.4.*

We also added the following sentence to the Conclusion:
*"The calculation of emission estimates from single flight transects is not advisable, because the single transect estimates showed deviations from their mean and the kriging method of more than 40% in both directions."*

**Grammar**
Line 43: change "affect" to "affecting". – *Done.*
Line 473: "don't" to "do not". – *Done.*

---

## Author Comment (AC2) · 24 Aug 2020

**Answers to reviewers for "Estimating CH4, CO2, and CO emissions from coal mining and industrial activities in the Upper Silesian Coal Basin using an aircraft-based mass balance approach"**

We would like to thank the two reviewers for the suggestions to improve the manuscript. Below you find our answers to their comments. The reviewer's comments are written in normal font, our answers in italics.

**Review 2 by Anna Karion:**

This is a well-written, well-researched manuscript and should be published in ACP after 2 major corrections noted below. Overall, the authors have done a lot of work on this flight data (although only 2 flights). But especially given that there are only two flights, there are two additional revisions that should be considered.

1) Characterization of the wind conditions prior to the flights. This is done qualitatively ("relatively steady"), but not quantitatively, and can be used in the sensitivity analysis and uncertainty calculation. The mass balance requires the wind to be constant (Eulerian equation). If it is not constant over the transit time, there is uncertainty.

*Thank you for this comment! It is true that the wind prior to the flight is very important for the mass balance. Here we are in the advantageous position of having wind profile measurements of the region for the entire day. We added the wind speed during the four hours prior to the downwind sampling to Table 3. As suggested, we added another sensitivity analysis to the uncertainty calculation, using the standard deviation of the wind lidar wind speed measurements in the four hours before the downwind sampling as temporal wind speed uncertainty estimate. We added a paragraph in Section 3.4 describing this sensitivity test and updated Figure 7 to include this uncertainty measure: "One assumption for a mass balance calculation is that the wind is constant during the time it takes for the gases to be transported from the emission source to the observation location. In reality the wind field is subject to considerable variability. In our case we were able to assess this temporal variability from the wind lidar observations. To account for wind variability, we calculated the standard deviation of wind speed within the boundary layer and added it to the kriged wind field used in the mass balance calculation. This introduced an uncertainty of 17% and 15% to the morning and afternoon flight results, respectively."*

*The uncertainty due to the wind variability, here in purple bars, is proportional to the wind speed uncertainty, because the wind speed is a linear factor in the mass balance equation. The addition of this sensitivity test increased the uncertainty of the mass balance estimates. The new uncertainty values have been updated throughout the manuscript.*

[Figure]

2) Use of biosphere model in sensitivity analysis. This is less important to the final result, but in the sensitivity analysis using the upwind transect as the background condition, the method of use of the biosphere model is confusing. The authors only subtract the biospheric influence one hour prior to measurement, because the upwind measurement was made one hour prior. This needs more explanation to me at least. However, since this method is only used to look at the sensitivity of background choice, it is a more minor issue than the above. More discussion on both these points is below, with other detailed/minor comments.

*The biosphere model VPRM calculates the contribution of biogenic sources to the $CO_2$ mole fractions in the atmosphere, by parameterizing the gross ecosystem exchange and ecosystem respiration fluxes. In our case we calculate contributions to downwind observations because when using an atmospheric background calculated from the upwind flight track, this biogenic uptake changes the background relative to the enhancements caused by anthropogenic $CO_2$ emissions. Ideally, a mass balance flight should sample the same air mass upwind and downwind of the emission sources. Then the contribution of the biogenic sink could be inferred from the trajectories between the upwind and downwind flight track as a sum over the entire transit time.*

*We acknowledge that by using the model in the manner described we have added a layer of complexity to a phenomenon already difficult to begin with. Using the model, in theory, is supposed to allow us to do more than just grossly estimate biospheric influence assuming heterogeneity in space and time - we also aimed to take into account its spatio-temporal variability. We believed that for our study they might have been of some importance, as the area has many fragmented forests intertwined with substantial urban- and rural-type areas.*

*As was mentioned above, the optimal use of the model in the method described would require for the upwind track to be flown in exactly Lagrangian manner. Specifically, this could be done by either 1) flying at the \*same upwind location\*, however four hours before the downwind measurements were taken, or 2) flying at the \*same time\* (i.e. as was in fact flown), but closer to the downwind track at the time (along the red line from Fig S2). Each of those options would allow then for a direct link between the simulated mole fractions and those measured aboard the aircraft, i.e. both values would be available at the same time and location, which would, in theory, minimize the errors caused by the inhomogeneities in the spatial distribution of the biospheric sources. Neither of those strategies was adopted, and even in case they were, we would still have to face other challenges: for scenario 1) measurements done earlier, during PBL development, tend to be poorly represented in the models, for scenario 2) the area constrained by the mass-balance method would not have encompassed the full study area – c.f. Fig S2.*

*We have thus decided to adopt a hybrid approach, in which we assume that we can still link the measurements to our model quasi-directly, despite the fact that the model results are simulated for a location several tens of kilometers away from the actual upwind measurement location. It should be noted that it is quietly assumed here, that the biospheric fluxes are spatially homogeneous, i.e. fluxes in the area between the downstream transect and the location of the trajectories one hour before are similar to fluxes in the area between the location of the trajectories one hour before and the upstream transect. This explanation has been added to Section 2.2.2.*

[Figure]

*In fact, the above figures show that there is very little difference: The left figure shows the biospheric signal accumulated along the 1 hour back-trajectories for each receptor point along the downwind transect at different heights (small diamonds, values between 0 ppm and -1.5 ppm) together with those accumulated along the full trajectories between upwind and downwind transect (values between -2 and -5 ppm). Note that almost all of the differences between these are related to the longer duration of the full trajectories of 3.5 hours on average. The right figure shows the differences between 1 h and full-period accumulations after normalization (i.e. the VPRM CO$_2$ contributions are divided by the duration of the trajectory). This clearly shows that there are very small differences in fluxes over the area along the 1 h trajectories and along the full trajectories connecting upwind and downwind transects.*

L23: "estimates... which are well within the range"... – *Done.*

L60: e.g. reads awkwardly, I would write "which are used in climate projections, for example." – *Done.*

L87 should say "using an airborne eddy covariance" (or maybe "the airborne eddy covariance..."). – *Done.*

Figure 1: Could there be an inset to show the larger map where this is located? The google earth image does not actually show any location boundaries etc, or lat/lon indicators. – *Done.*

L102: make and model or reference for "well-established" CRDS? Were all 3 gases measured by both instruments? [I see now this is further described in Section 2, so perhaps just refer here to that section].
*We removed "...using a well-established cavity ring-down spectrometer (CRDS) and a modified quantum and interband cascade laser spectrometer (QCLS, Kostinek et al., 2019)." Since this is elaborated in Section 2.*

L108 - this is a very nice aspect of this study that is usually not done! – *Thanks!*

L112: Perhaps Chapters should be "Sections"? (this is up to the editors). Later the text does refer to "Sections", so consistency would be good. – *Done.*

L124: Earlier several remote sensing instruments were mentioned, but this work only focuses on the in-situ measurements. Is there a reason for this?

*During CoMet a wide range of instruments were employed. Each of the methods delivered individual results that will all be published within the CoMet Special Issue. Including remote sensing into this study would have significantly enlarged the manuscript since there are several issues that need to be addressed. Nevertheless, a comparison of in situ and remote sensing observations is also planned.*

L138 I don't think there should be a comma after both. – *We deleted this.*

L165-166: The wind speed (and direction) also must remain constant over the transport time. This is a key assumption that the flux measured out of the downwind plane (or "wall") is equal to the emissions flux from the surface. Variability from this assumption is likely to happen, so that should be accounted for in the uncertainty analysis. Using wind speeds measured by the aircraft in the afternoon may not be the correct approach if they are not representative of the wind field over the whole domain over the transit time. If downwind wind is used, it should be shown that the wind speed was constant (or what the variability was) through the time it took for the air to transit the domain. Fig 6(a) in Karion et al., ACP, 2019 (https://doi.org/10.5194/acp-19-2561-2019) shows (granted, in an extreme case with long residence time) the difference between true emissions and measured emissions in a downwind plane using a forward model. Given the relatively small domain here, it's likely fine to assume steady winds over transport time, but this should be stated (and the variability of the wind included in uncertainty/sensitivity analysis).

*We included this uncertainty. See our comment above.*

L211, using the upwind as a "check" has been done before, citing some literature here would strengthen the justification of this choice.

*We added another sentence on the upwind transect and a reference in the manuscript.*

L229 "form" should be "from" – *Done.*

Section 2.2.2. It is not clear to me why only the last hour of the footprint is used to estimate influence of biogenic fluxes on the downwind transect. For the final mass balance, the edges of the plume are used as the background, so I don't understand why the time between the upwind and downwind sampling matters. For the case of using the upwind transect, I am still not clear on the use of the 1 hour time frame. More on this in the next section.

*We added a clarification that the biogenic uptake is only necessary for the use of an upwind background mole fraction. Please also see our answer to your general comment on this matter.*

L264: Very nice detailed error analysis: This is where I believe that you should include the variability of the wind over the transport time, not just measurement error. Perhaps added in quadrature to the 0.3 m/s, and constant over the whole grid. Or, now reading 2.3.2, perhaps it should be considered in the sensitivity portion, as it would be a systematic error.

*We included it as a sensitivity test.*

L283: Table 1 repeated. – *Corrected.*

L328: no comma after 'cases'. – *Corrected.*

L388: Agree! This addresses my issue above about steady winds over the transit time - so the section 2.2 above should mention this as well. In table 3, I would recommend also showing the mean wind and standard deviation for the Lidar over the 3-4 hours transit time, not only the flight time (to address my earlier comment). Several times this section says "relatively steady" - using the lidar data, this can be quantified and stated here.

*Please see our answer above.*

L429-435: This is one issue with Kriging in space samples that were conducted at different times! It seems that hopefully the correction did the job, but in general, one might think this is a reason to do separate mass balance

calculations for each transect (with each its own PBL depth), and then average. But either way, changing conditions are difficult to deal with, and the authors have made a decent attempt at accounting for this, so no need to do this.
*We now added an extra-study where we used the individual tracks using the same PBLH for all of them. Results are listed in the new Section 3.5.*

Figure 5: These are nice useful figures. What is the averaging time prior to Kriging? Or is it the native 0.5-Hz data that is Kriged?
*The averaging time prior to Kriging is 20 seconds. This is stated in the method section 2.2*

L490. To clarify the method for using VPRM above, is it not accounted for at all in the version where the edges are used as background, and only when the upwind background is used? – *Correct, the VPRM model is only used for the upwind background.*
This makes sense if the assumption is that the uptake is the same at the edges and in the center, which in this case is a reasonable assumption (maybe not in an urban area). Does the enhancement from VPRM not change across the transect? That would then justify not including it when using the edges as background.

*The VPRM implementation used in our study did, in fact, take into the account the variable land-use and vegetation types (including also urban areas) in the area of influence of our observation. However, the resulting simulated uptake signal only shows a small trend along the transects as can be seen in this figure. Thus, we calculated the downwind background for both edges and used an average of these values to account for the gradient.*
*Please also note our answer to the general comment above.*

Regarding the upwind background use of VPRM: I am still confused by the use of one hour only, and given how variable the uptake is in time due to the diurnal cycle of fluxes, it's not clear how to deal with this. If biogenic fluxes were constant in time and space, then the full trajectories (footprints) should be used until the location of the upwind leg (…).
*This is not exact. Even if the fluxes were constant in space and time, then applying VPRM over the trajectory connecting upwind and downwind points would still simulate a too large uptake. To visualize this, we can imagine our study area as a part of an infinite, perfectly flat and perfectly homogeneous forest. Over such an area the mole fractions of $CO_2$ would be independent of space (as much as stochastic mixing allows, naturally) – but would still retain its time-dependence, as the diurnal biospheric cycle still works. Doing the same measurements of mole fractions over our flight tracks, we would still have a single hour of temporal difference in observations, and four-hour difference in the air-mass flow between measurement locations, during which the biosphere was able to uptake more $CO_2$. In such a scenario, if we would use the model and predict the change of mole fraction as suggested by the reviewer, we would have simulated a drop of mole fraction that was caused by four hours of uptake, which is incompatible with the measurements that we did. In this case we should again compare against influence simulated from a single hour. The simulated mole fractions would perfectly fit the observations thanks to spatial homogeneity.*
*However, in more realistic scenarios of inhomogeneous biospheric fluxes, the spatial decoupling between observations and simulated single-hour trajectories will cause errors that might be more challenging to quantify. This is in fact the situation that we are dealing with here – see the discussion and figure under the general comment above.*

(*continued*) Otherwise, I think this is too complex to handle in the method it has been handled here. If the upwind leg were flown one hour *after* the downwind legs then how would this work? I am wondering if the upwind transect just cannot be used other than to show the lack of significant upwind fossil sources. Please explain/justify the use of the one hour time frame, and how this would work if the upwind leg were flown simultaneously or after the downwind leg - would you not subtract VPRM at all in that case? Seems wrong.
*In case the upwind track would have been flown one hour after the downwind sampling, the $CO_2$ background mole fractions in the boundary layer would have further decreased due to continued biogenic uptake. In this case the upwind background observations would be even lower than the downwind background and would need*

*to be corrected with one hour of biogenic uptake. Using the model to estimate this correction becomes meaningless, as we cannot then establish a link between the downwind and upwind mole fractions using our model framework.*

L515: Definitely agree with this statement on biosphere-atmosphere fluxes being complex! – *Thanks.*

L520 (sensitivity using average wind speed during downwind legs): This is close to what I suggested earlier. However, this is the average over the flight time, not transport time. One could use the transit time instead (from lidar perhaps, or from the model), as was done by Karion 2013 & 2015, to see the effect. What matters more is what the wind was at the location and time of emission - if it was high, then less $CH_4$ was picked up in that air mass, if low, vice-versa.
*We added a sensitivity test using wind variability during the transit time.*

SI: Table 1 can the caption also explain the offset of local time from UTC so the reader can quickly translate the UTC hours to local? – *Done.*

How frequent is "frequently calibrated" (approximately - daily? hourly?). Text S1 gives the mole fractions of the cylinders in ppm for both gases, but the text says ppm/ppb. Should be ppm/ppm.
The drift with time uncertainty is estimated using the flight time, so presumably the calibrations did not occur during flight, so frequently is > 2.5 hours?
*We calibrated after every second flight. This has been added to the text.*